# Surface Phenomena at the Interface between Silicon Carbide and Iron Alloy

**DOI:** 10.3390/ma14226762

**Published:** 2021-11-10

**Authors:** Mirosław Cholewa, Tomasz Wróbel, Czesław Baron, Marcin Morys

**Affiliations:** Department of Foundry Engineering, Silesian University of Technology, 7 Towarowa Street, 44-100 Gliwice, Poland; miroslaw.cholewa@polsl.pl (M.C.); czeslaw.baron@polsl.pl (C.B.); marcin.morys@gmail.com (M.M.)

**Keywords:** casting composite, silicon carbide, gray cast iron, graphite, pearlite, reinforcement particles, metallic matrix

## Abstract

The paper discusses a potential composite produced using the casting method, where the matrix is gray cast iron with flake graphite. The reinforcement is provided by granular carborundum (β-SiC). The article presents model studies aimed at identifying the phenomena at the contact boundary resulting from the interaction of the liquid matrix with solid reinforcement particles. The scope of the research included, primarily, the metallographic analysis of the microstructure of the resulting composite, carried out by using light (LOM) and scanning electron (SEM) microscopy with energy dispersive X-ray spectroscopy (EDS) analysis. The occurrence of metallic phases in the boundary zone was indicated, the contents and morphology of which can be optimized in order to achieve favorable functional properties, mainly the tribological properties of the composite. In addition, the results obtained confirm the possibility of producing similar composites based on selected iron alloys.

## 1. Introduction

The concept of the paper and direction of the conducted research is based on the use of gray cast iron as a matrix and silicon carbide (SiC) particles as reinforcement. Ultimately, the aim is to obtain a material resistant to frictional wear with high coefficient of friction. A hypothetically hard reinforcement phase promotes an increase in wear resistance. Graphite present in the matrix microstructure can act as a solid lubricant, and thus, it can stabilize the frictional conditions, providing the ability to damp mechanical vibrations, typical for cast iron. In addition, graphite in cast iron increases thermal conductivity compared to iron alloys without the graphite phase. It is predicted that the potential applications of the composite produced as part of the paper include brake discs, as well as working parts of mining and construction machines. The condition for further operational research is to obtain favorable micro- and macrostructural features of the presented composite. In addition, it should be initially noted that the analysis of surface phenomena presented in the paper will become the basis for further structural and mechanical studies of the innovative composite gray cast iron—SiC.

Proper understanding of surface phenomena at the interface between ceramics and metal matrix is the basis for designing new materials—e.g., showing the properties of composites. The widely tested and used composites have a matrix of light Al and Mg alloys, which can be obviously justified—these are materials with favorable performance properties—in this case showing minimum mass density and maximum mechanical properties.

The subject of the study is a slightly different association of potential components. They include silicon carbide in the allotropic β form, i.e., not completely pure, relatively cheap and available, and an iron alloy similar to cast iron with flake graphite. The metallic matrix is a cheap, well-known material with versatile functional properties and the lowest processing temperature of iron alloys.

The subject of the research is therefore the combination of materials with possibly minimally differentiated carbon concentration in the ceramic–metallic matrix system. Here, attention should be paid to the driving force of the diffusion process. One of the several important factors is the level of the concentration difference at the component boundary. Carbon, having a small atomic radius, is easily diffusible. In addition, silicon carbide belongs to ceramics featuring maximum thermal conductivity and very high hardness, which bodes well, for example, when creating potential new wear-resistant composite materials. According to the adopted concept, silicon carbide remains in contact with the liquid iron alloy. Such a system is highly chemically reactive. In many definitions of composites, chemical reactions between components are disqualifying in terms of a classical composite. Nevertheless, the indicated solution seems to be attractive in terms of technology and applications provided. The intention is to control the physicochemical reactions and create a favorable combination of ceramics with metal in order to manufacture a good material, perhaps even only showing the characteristics of a composite.

The above-mentioned concept components constitute the content of the paper. The phenomena at the interface between the components are considered: cast iron matrix–technical SiC as reinforcement. It is expected to form a material showing potentially high wear resistance, based on cheap and well-known engineering materials and using foundry technology.

Therefore, the aim of the study is to check the possibility of creating a material with composite properties based on components showing physicochemical reactions.

During chemical reactions SiC is decomposed, while physical reactions consist in forming solutions and phases with the components of the matrix alloy. The control and steering of the reactions should ensure their stopping in a timely manner—so as to keep the ceramic particles in the solidified matrix. Foundry processes provide similar conditions, additionally allowing to control crystallization processes to optimize the microstructure particularly in the interface of the components.

The use of silicon carbide in the metallurgy and foundry of ferrous alloys is quite common and fundamentally different from the concept presented in the paper. In particular, SiC is used effectively during the melting of gray cast irons of any form of graphite—from flake to nodular form. Its acts in two ways. It increases the content of carbon and silicon in the alloy and modifies the microstructure, thus improving the functional properties of cast irons. In terms of technology, it is an excellent alternative to commonly used carburizers in the form of e.g., coal coke, petroleum coke, anthracite or natural and synthetic graphites. Increasing the silicon content in the alloys is generally carried out by adding ferrosilicon (FeSi). In many cases, it is necessary to apply both metallurgical treatments to the liquid alloy. Moreover, both treatments have a significant impact on the microstructural features of cast irons. They have a positive effect on the shape, number and dispersion of the precipitated particles of graphite. They also affect the distance between the ferrite and cementite plates in the pearlite. The treatments of metallurgical modification optimize the microstructure of cast irons. This is generally achieved by increasing the number of nuclei [1,2,3,4]. It is the silicon derived from SiC that has a modifying effect, protecting the alloy from excessive crystallization of cementite (Fe_3_C), which limits the crystallization of graphite, increasing the hardness and brittleness of cast irons—usually locally in areas with high cooling rate. Considering the above, SiC is an excellent component in metallurgical processes for the production of cast iron, providing the desired microstructure and reducing the sensitivity of cast iron to the cooling rate that varies across the section of castings with different wall thicknesses. In the case under consideration, this makes an important advantage, because fragmentation/modification of microstructure directly at the interface between the components of composite is to be expected.

It seems that the modification treatments with SiC are gaining importance for the so-called synthetic cast iron obtained e.g., by remelting steel scrap with low carbon content. Graphite morphology may be disadvantageous when using some carburizers. In such cases, SiC has a modifying effect. An important advantage of SiC is that modification effect is extended in time—greater stability of the crystallization nuclei, which is probably due to the longer dissolution process compared to the alternative use of ferrosilicon (FeSi). The mechanism of action is hypothetical. The formation of graphite clusters around SiC particles was observed. This is the result of the local supersaturation of the bath with carbon and silicon. Eutectic graphite clusters can be also found in the presence of ferrosilicon (FeSi). However, their stability over time lowers [2]. It can also be hypothetically assumed that the different diffusion of Si and C is not only local in nature, but also temporary and periodic. Similar phenomena can be minimized by introducing SiC particles with the minimum technologically achievable dimensions. The trend is obvious: maximization of the contact surface of the reactants, maximization of dispersion (also in purely mathematical terms), high reaction rate in the volume of the metal. The limitations include: a high degree of homogenization, i.e., intensive mixing—e.g., in an arc furnace or a dedicated induction furnace, and probably a relative shortening of the modification effect. An example of an attempt to solve the problem is the procedure described in the patent [5], where nanoparticles bound to microparticles can effectively act as a nucleator. The object is a nucleating agent for casting ductile or vermicular cast iron using silicon nanocarbon. According to the patent, the mixture of nano- and microscale SiC powder comprises nano-SiC particles attached to the surface of micrometer-sized SiC particles.

Critically speaking, the information provided about the reactions between the key components—the reactants, are obvious, and can be controlled by many factors. They include temperature, component shares, granularity and time. Such a state opens up many technological possibilities of producing a technologically advantageous and useful material containing SiC in a matrix containing Fe.

Considering the composite production, the solutions presented above can be used for melting the matrix in form of cast iron with the use of specialized treatments, also for melting “synthetic” cast iron from steel scrap.

Despite the fact that SiC is thermodynamically stable, it is subject to oxidation resulting in covering with SiO_2_ oxide on the surface. Among the reinforcement materials in composites, this surface phase (SiO_2_) in some cases, in general, the oxide phase may facilitate wetting and affect the microstructure of the transition zone depending on the metallic matrix of the composite used. Composites based on alloys, e.g., Al with SiC particles, are commonly known and used. Composites with dispersion reinforcement in the form of SiC microparticles dominate here. It is common to try to minimize their size, inter alia, in order to limit the gravitational segregation of components. This can be carried out by increasing the share of internal friction forces and viscosity, which depend on the particle development area. The phases of the transition zone determine the site of proper composite formation. It is theoretically and practically the most sensitive microarea of each composite.

With large particles, the oxide surface represents a relatively smaller share and its possible removal should be easier. Slag-forming materials or dedicated surfactants can thus facilitate the production of the composite in the liquid state. The controlled process of crystallization is aimed at providing proper modification—generally the fragmentation of the microstructure in the boundary zone of the components.

The presented system of components (FeC–SiC) basically meets all theoretical assumptions for the production of a correct composite in the liquid and solid state, with the controlled primary crystallization of the matrix and phases in the transition zone. It is advisable to provide heat treatment, despite a significant difference in thermal expansion of the components.

The available literature shows few similar concepts of MMC_p_ (Fe–SiC) composite production and significantly diversified in terms of technology and practical applications.

For example, the composite material [6] produced from silicon carbide and iron has high strength, hardness, wear resistance, heat resistance and oxidation resistance. The method comprises the following steps: silicon carbide and iron powder are mixed and evenly ground to obtain a homogeneous sample, then the material is formed using a hydraulic press and the sample is coated with iron powder and then sintered. Thus, the process consists of multiple stages and does not apply to the casting techniques.

Another similar solution in the field of sintering is the composite with the ferrosilicon phase in situ [7] obtained by using the carbothermal reduction method, thanks to which a porous composite material SiC–Fe–Si can be produced. The ingredients include: 35–73 %wt. silicon carbide, 5–12 %wt. carbon powder and 22–53 %wt. iron oxide; the added binder weight ranges from 1 to 5 %wt. of the total weight of the raw materials. The composite is produced by mixing, pressing and molding, as well as sintering in argon. As in the previous case, the process consists of multiple stages and the obtained material is porous, which, in principle, leads to its mechanical weakening compared to monolithic materials.

Another example of creating a similar composite in terms of materials used refers to a composite material based on iron and aluminum, enriched with silicon carbide [8]. The product has good high temperature strength and nonoxidation properties, as well as a low thermal expansion coefficient. The silicon carbide reinforcement is 0.1–1.0 %wt. of the total weight and the particle size ranges from 1 to 5 nm. The manufacturing method comprises the following steps: pretreating a silicon carbide reinforcement, melting, homogenizing annealing, forging, hot rolling, wire drawing and heat treatment. Thus, the intended use and manufacturing technology, despite the fact that they differ from the presented concept, indicate alternative applications and manufacturing possibilities. Furthermore, they indicate potential directions for development by alloying the composite with aluminum.

Potentially, the chemical components of the matrix can perfectly regulate surface phenomena. The material quoted below was also made by using sintering techniques with reinforcement in the form of SiC particles. This is an Fe-based composite material reinforced with SiC particles [9]. The matrix material contains the following components in percent by weight: 3.4–3.6 %wt. C, 1.9–2.1 %wt. Si, 0.3–0.5 %wt. Mn, 0.5–0.8 %wt. Cr, 0.1–0.2 %wt. Ti, 0.05–0.10 %wt. Te, 0.03–0.05 %wt. Mg, 0.01–0.03 %wt. Re, at most 0.05 %wt. P, at most 0.02 %wt. S and balance Fe. According to the authors, silicon carbide particles can be well combined with the substrate to effectively improve the mechanical properties of the material subjected to sintering and quenching. After quenching and tempering heat treatment from 1250 °C, the maximum hardness and tensile strength can reach 45.4 HRC and 1859 MPa. Because of its good mechanical properties, its practical applications relate to specialized technical applications that are not necessarily common, including tribological applications.

The composite presented in [10], made by using an iron alloy containing the following components in percent by weight: 1.5–5.5 %wt. Al, 3.6–7.0 %wt. Mo, 2.0–3.3 %wt. Cu, 0.2–1.5 %wt. Cr, 12.0–22.0 %wt. Zr, 2.5–4.5 %wt. Mg, 1.0–2.0 %wt. Mn, 1.2–1.4 %wt. Si, 0.08–0.14 %wt. B and SiC whiskers, is conceptually most similar to the one presented in this paper. Whiskers as the main hard phase can be well combined with matrix, thus effectively improving the mechanical properties of the sintered and quenched material. After the tempering and heat treatment is carried out, the hardness and strength of the alloy increase significantly. There is no doubt that similar materials can be used in kinematic pairs exposed to abrasive wear. However, the process still bypasses casting techniques, ensuring freedom of shape and minimizing production procedures.

On the one hand, the presented examples show the high potential of Fe–SiC material association, while on the other hand, they show specialized and precisely dedicated technologies. The examples should be a rationale for the effective use of components that are available and technologically friendly. By design, the presented references are not substantially theoretically underpinned, but with the knowledge available, they can focus many research projects on similar areas.

The practical combination of cast iron with flake graphite, together with silicon carbide, places this material in the category of wear-resistant materials. A typical combination of hard and soft phases should provide the effect of minimal wear even in conditions of little or no technical lubrication. The soft graphite phase has a lubricating effect and makes a storehouse for wear products. The hard SiC phase increases the wear resistance, and the ferritic-pearlitic matrix transfers loads mainly compressive in nature. Compared to typical composites, e.g., on the Al matrix, the suggested material has a higher mass density, but also a hypothetically greater potential for frictional wear under many different conditions of cooperation of kinematic pairs, e.g., under conditions of erosive or abrasive wear.

Overall, the technology of casting a composite on a cast iron matrix with SiC particles may be a favorable solution in terms of simplicity of the technology and the attractiveness of tribological practical applications. The prerequisite is to obtain the correct surface phenomena at the interface of the components, which was adopted as the main aim of the study.

## 2. Materials and Methods

This research project presents a model that shows the matrix of the composite in the form of the simplest gray cast irons with flake graphite, approximately eutectic with a typical saturation factor and eutectic equivalent. The research ignores the influence of typical alloy components increasing mechanical properties, in particular tribological ones, e.g., carbide forming or improving castability and lowering the matrix melt viscosity. The purpose of their use may be to increase the hardness of the metallic matrix and to facilitate the wetting processes of SiC particles. Apart from Fe and C, this type of component includes Si, Cr, Mo and V.

The second component, i.e., the composite reinforcement, is SiC, which is not as obvious as the matrix material. In classic production processes, there are three classes of commercial SiCs differing in the level of thermodynamic stability. Conditionally thermally stable α-SiC, thermally unstable β-SiC and reactive metallurgical SiC. β-SiC is black in color and contains more α-SiC than the green variant that has such impurities as Fe_2_O_3_, Al_2_O_3_, CaO, SiO_2_, MnO_2_ (the terms “black”, “green” and “metallurgical” are trade names). The chemical composition differs depending on the producer, and an example of the differentiation of the composition of individual varieties is given in Table 1.

Silicon carbide together with boron carbide (B_4_C) are covalent carbides. The tetrahedral system of covalent bonds enforces the tetrahedral ordering of Si and C atoms. Thus, their typical structures are the analogs of sphalerite β-SiC and wurtzite α-SiC (according to the Somerfeld rule). The β form is considered to be unstable in the entire temperature range, but it most often crystallizes under conditions where the primary synthesis product does not recrystallize [12]. In the literature [13], over 100 SiC polytypes have been described. The differences of polytypes are manifested by the different sizes of unit cells. The causes of polytypism have not yet been clearly explained.

From among the available SiC varieties, the thermally unstable β-SiC was selected for testing, most often occurring in one basic structure of 3C sphalerite, counting on the possibility of controlling physicochemical reactions at the contact boundary: Fe–(β-SiC) alloy. Confirmation of the possibility of obtaining limited reactions is the not fully satisfied stoichiometry requirement. It is expressed by the molar ratio Si:C = 1.03:1.05. Thanks to this, SiC undergoes slow thermal decomposition, and the resilience of the decomposition products reach measurable values, starting from the temperature of 1500 °C [14].

Summing up, the tests assumed gray cast iron with flake graphite as a metallic matrix, with the chemical composition presented in Table 2 and determined using the LECO GDS500A (LECO Corporation, St. Joseph, MI, USA) emission spectrometer. On the other hand, the reinforcement is β-SiC with a significant model particle size, facilitating the wetting of the components, with a significant reserve for physicochemical reactions reducing the volume of the reinforcement particles. The β-SiC particles with a grain size of 0.8–1.6 mm were used, while the Fe_met_ content presented in Table 3 in fact represents the presence of iron oxide range from FeO to Fe_2_O_3_. In general, Fe_2_O_3_ oxide is the dominant one.

The composite subjected to testing was produced by casting technology. Liquid gray cast iron at a temperature of 1530 °C was poured by gravity into a sand mold with a granular preform β-SiC placed in its cavity. Then, microscopic metallographic tests were carried out on samples etched with Nital with the following composition: 5 mL HNO_3_ + 95 mL C_2_H_5_OH using the NIKON ECLIPSE LV150N (NIKON Metrology Europe NV, Leuven, Belgium) as OM (optical microscope) and on nonetched samples using the Phenom ProX (Phenome-World, Eindhoven, The Netherlands) as SEM (scanning electron microscope) with BSE (backscattered electron) imaging, 10 kV electron beam accelerating voltage and EDS (energy dispersive spectroscopy) analysis.

## 3. Research Results

An example of a cross-section of a gray cast iron–silicon carbide composite produced by the casting method is shown in Figure 1.

On the other hand, Figure 2, Figure 3, Figure 4, Figure 5, Figure 6, Figure 7 and Table 4 show the results of metallographic tests. It was found that the obtained chemical composition of the cast iron matrix and its crystallization conditions in the sand form determined the final microstructure consisting of flake graphite in a pearlitic matrix (Figure 2). In addition to the above-mentioned phases, the microstructure of the tested gray iron contained a small amount of steadite (phosphorus eutectic) and nonmetallic inclusions in the form of MnS sulfides.

In the area of contact between the cast iron matrix with the particle reinforcement β-SiC, the dispersion of flake graphite grew. The area of the fragmentation of graphite extended into the matrix by a distance calculated in fractions of a millimeter (Figure 3a). In addition, a significant ability of cast iron to infiltrate into the spaces between β-SiC particles was observed. Figure 3b shows the gap filled with a metallic matrix with a thickness of tens of μm. This effect confirms that the flowability of the cast iron used was particularly high. That was probably caused by a local increase in the concentration of C and Si at the front of the metal stream as a result of the component dissolution reaction in the contact of the matrix with the reinforcement. Certain confirmation of the above statement comes from the results of the microanalysis of the chemical composition in point 2 in Figure 4, presented in the form of a spectrogram in Figure 5b and in Table 4. By contrast, Figure 3c shows the transition zone locally occurring in the area of the matrix contact with the reinforcement, which should be classified as a group of nonwall crystals. Its location was typical for this type of phases occurring in in situ composites. They show a significant diversity of chemical composition and a high degree of fragmentation (Figure 6). An example of the chemical composition of the transition zone, i.e., point 3 in Figure 4, is presented in the form of a spectrogram in Figure 5c and Table 4. In fact, this is not a chemical compound, but rather a phase formed at the interface between physical and chemical reactions. Precipitations of this phase appeared in the places where the reactions occurred the earliest, and diffusion into the matrix dis not degrade it. The chemical elements appearing in this phase included Fe, Si, and C in descending order, while its structural components were made up of very fine graphite in a metallic matrix (Figure 6 and Figure 7).

The transition zone under analysis, due to the phase composition, had mechanical properties located between the ceramic reinforcement and the cast iron matrix. This phase increased at the expense of the β-SiC particles, manifested by its rounded contact surface with the reinforcement particles. This surface was different from the raw β-SiC surface shaped by mechanical crushing. With a favorable and optimal proportion of morphology and dispersion, this type of transition phase may minimize the effects of the differential expansion of the two components of the composite and thus may affect its thermal properties. The differentiation of the degree of fragmentation of the flake graphite (Figure 6) indicates a locally variable intensity of the reaction. In addition to the effect of different wetting processes, this may also suggest a different reactivity of the β-SiC particle surface, which may also be justified by different specific surface, edge and tip energies of silicon carbide crystals forming polycrystalline β-SiC particles.

The second phase of the transition zone, the chemical composition of which, i.e., point 4 in Figure 4, is shown as a spectrogram in Figure 5d and Table 4, probably composed of iron and silicon oxide with an admixture of carbon. The presence of carbon in a significant share (even with the error of the EDS measurement method) excludes the existence of Fe_2_SiO_4_ fayalite at the moment of capture. This was the moment when the reaction was stopped due to cooling and crystallization of the system.

## 4. Discussion of Research Results

Based on the obtained research results, the following hypothesis can be formulated on the creation of a connection at the boundary of the contact between the cast iron matrix and the silicon carbide particles reinforcing the composite in the casting technology used. The oxidized surface of the β-SiC particles consists mainly of SiO_2_. There are two compounds in the composition of silicon carbide in the amount of up to approx. 1%wt. SiO_2_ and Fe_2_O_3_. However, in the metal bath there are iron oxides with a significant share of Fe_2_O_3_. This is due to classic metallurgical treatments. Research by Naro [16] showed that Fe_2_O_3_ can react with SiO_2_ silica to form iron silicate fayalite (Fe_2_SiO_4_) that makes a physical barrier preventing the dissolution of gases in the metal. In the liquid alloy β-SiC is slowly dissolved, which is a condition for the deoxidation effect [17]. In principle, these phenomena do not apply to the case under consideration, but they should be taken into account, as their occurrence is likely at conditions above 1500 °C, i.e., at the temperature of pouring liquid cast iron into a sand mold with a β-SiC preform placed in its cavity. Chemically bound silicon on the surface of β-SiC particles stimulates subsurface enrichment of the particles with carbon, which, based on the difference in concentration, diffuses into the metallic matrix in the immediate vicinity of the silicon carbide particles. Hypothetically, the sequence of chemical reactions is as follows:2Fe_2_O_3_ + 2SiO_2_ → 2Fe_2_SiO_4_ + O_2_(1)
2FeO + SiO_2_ → Fe_2_SiO_4_(2)
3Fe_2_O_3_ + 2SiC → 4Fe + SiO_2_ + Fe_2_SiO_4_ + CO + CO_2_(3)
3Fe_2_SiO_4_ + 2SiC → 6Fe + 5SiO_2_ + 2CO(4)
FeO + SiC → Fe + Si + CO(5)
3FeO + SiC → 3Fe + SiO_2_ + CO(6)

It should be emphasized that the effect of Fe_3_O_4_ oxide is generally ignored [16,17]. Probably, fayalite protects locally against physical migration of gases, so the released oxygen O_2_ promotes the oxidation of another surface of exposed silicon carbide. The periodicity of the reactions and their local character seem to be justified, as they significantly depend on the temperature field and thermal conductivity of the composite components in the microarea “reinforcement particle–matrix–environment”. The remaining products are Si and CO. Silicon has a local modifying effect, while CO along with the temperature decrease, in accordance with the Boudoudard reaction (C + CO_2_ ↔ 2CO), is released with graphite “in situ”, which may have a nucleating effect on its crystallization in a metallic matrix, i.e., ferritic, pearlitic or ferritic-pearlitic. Silicon dioxide together with fayalite forming physical solutions should constitute the phase components in the transition zone between the components. With a sufficiently high volume, they can segregate and constitute slag-forming components.

In addition, carburization of the matrix may locally lower its liquidus temperature. This in turn helps to lower the melt viscosity of the matrix, facilitating the wetting of the β-SiC particles. This means that the physical dissolution process is not instantaneous but progresses with the progressive thermal decomposition of β-SiC. By regulating the temperature and pouring time in the conditions of gravity casting, it is possible to stop the reaction by keeping the “frozen” β-SiC particles in the metallic matrix. Permanent active reactivity perfectly improves the conditions for creating a composite connection at the boundary of the contact of both components. In addition, graphite and pearlite are subject to modification in the contact zone of both components, i.e., in the most sensitive zone of each composite. The modification consists in refining and increasing the dispersion as well as improving the morphology of graphite and refining pearlite, i.e., reducing the distance between ferrite and cementite, while maintaining the thickness ratio in the proportion of 1:7. Such a system makes it possible to maximize the share of reinforcement in the metallic matrix well above the classic few up to dozen %.

## 5. Conclusions

By analyzing the results of the conducted research, the following substantive and application conclusions were formulated. The authors of the paper also presented the assumptions and directions of future research in the field of technological usability of the produced composite in configuration the gray cast iron matrix reinforced with SiC particles from the point of view of its mechanical properties.

### 5.1. Substantive Conclusions

The transitional phases at the contact point of the cast iron matrix with the silicon carbide particles constituting the reinforcement confirm the favorable wetting phenomena, conditioning the obtaining of a technologically useful composite.The type and morphology as well as phase dispersion in the transition zone require optimization depending on the intended use of the products made of gray cast iron–β-SiC composite. Typical foundry technological factors (casting parameters) make it possible to regulate the thermal capacity of the components, the initial conditions as well as the cooling and crystallization rate of the entire composite. Selection of technological factors, i.e., the matrix of the molding sand, as well as the initial temperature and mass of mold, depends on the massiveness of the casting expressed, e.g., by the casting solidification modulus or individual thermal centers in the casting, according to commonly used standards.Important for the quality of the resulting composite is the anaerobic metallic phase identified as very fine graphite in the metallic matrix. This phase, containing Fe, Si and C in its chemical composition, may be responsible for the fragmentation of the microstructure of the cast iron matrix in the vicinity of the β-SiC particles, and thus may act as a graphite modification precursor. The differentiation of the degree of fragmentation of graphite in this phase indicates the local variability of the rate of physicochemical reactions occurring between the components. In addition, the research results confirm the possibility of controlling the speed of these reactions. The reaction rate is affected by the temperature drop in the mold in the range from the pouring temperature to the solidus temperature. The diffusion phenomena in the liquid state of the matrix are the strongest.

### 5.2. Application Conclusions

Taking into account the mechanical properties of the gray iron matrix and the reinforcement in the form of β-SiC particles, it is possible to obtain the final material with the properties of an abrasion-resistant composite. The practical verification of this statement should be carried out on a test stand that faithfully reproduces the type of wear depending on the working conditions of the composite machine part.It is anticipated that the potential applications of the composite developed are working components of mining machines for the mining industry or for brake discs. The accepted hypothesis assumes that the use of the gray cast iron–silicon carbide composite for brake discs will allow us to obtain functional properties located between the commonly used materials, traditionally cast entirely of gray cast iron, and the special materials based on SiC ceramics.

## Figures and Tables

**Figure 1 materials-14-06762-f001:**
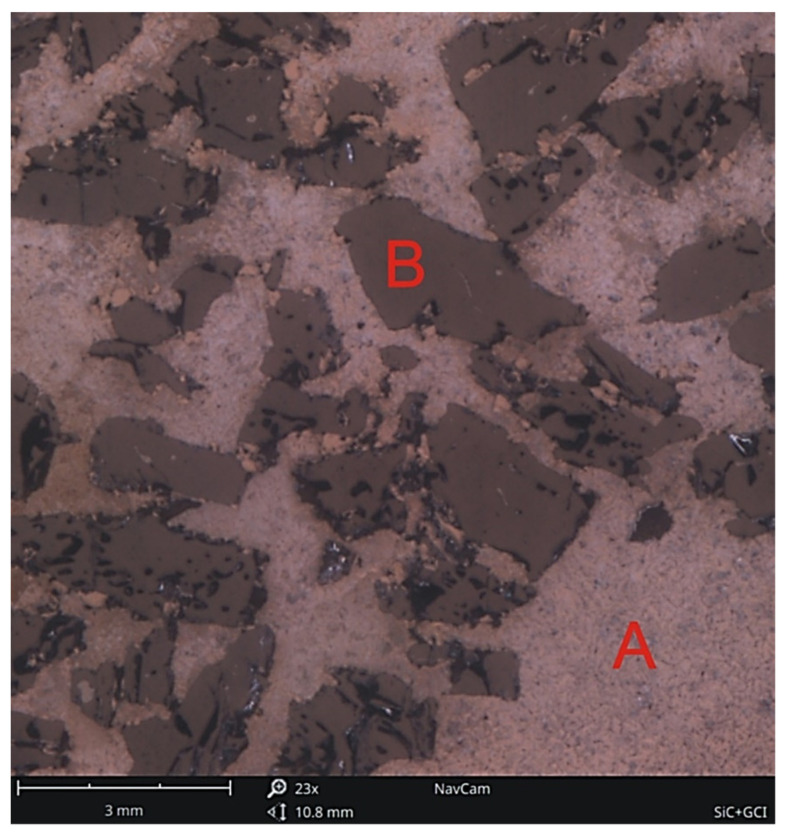
An example of a composite macrostructure of gray cast iron (**A**) and silicon carbide β-SiC (**B**), mag. 23×.

**Figure 2 materials-14-06762-f002:**
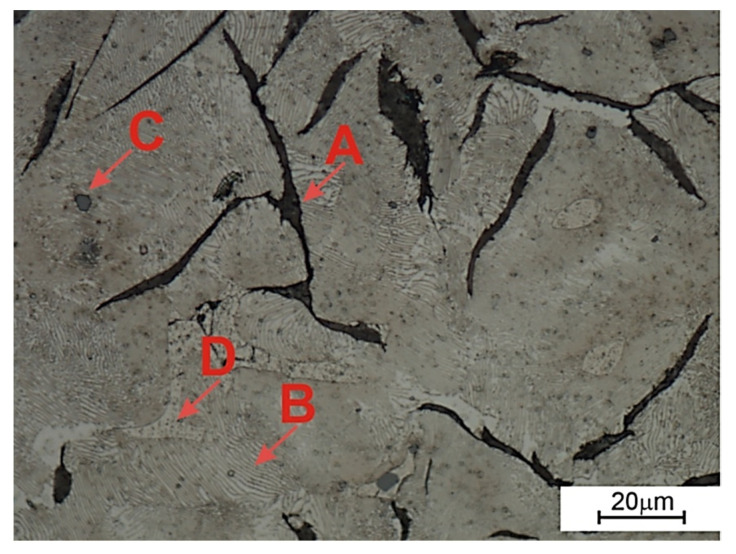
The microstructure of gray cast iron constituting the matrix of the composite, OM, mag. 500×, etching: Nital. (**A**) flake graphite; (**B**) pearlite; (**C**) MnS sulfides; (**D**) steadite.

**Figure 3 materials-14-06762-f003:**
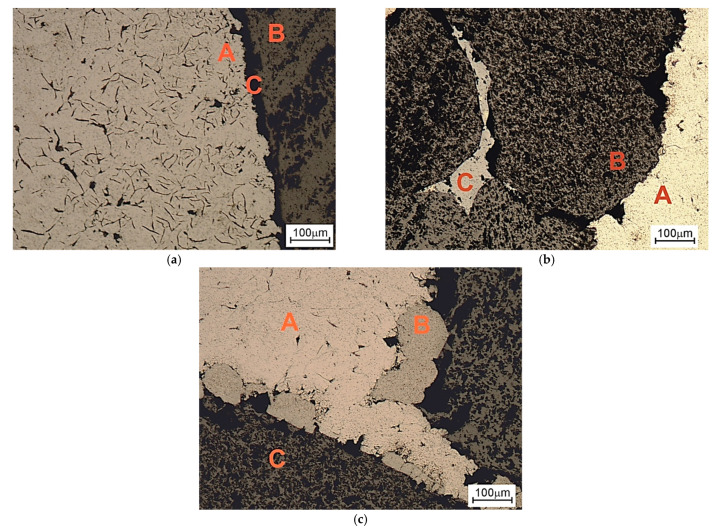
Composite microstructure of gray cast iron–β-SiC in the area of the matrix contact with the reinforcement, OM, mag. 100×, etching: Nital. (**a**) A—Fragmented graphite zone; B—β-SiC reinforcement; C—transition zone 2; (**b**) A—metallic matrix (gray cast iron); B—β-SiC reinforcement; C—gap filled with a metallic matrix; (**c**) A—metallic matrix (gray cast iron); B—transition zone 1; C—β-SiC reinforcement.

**Figure 4 materials-14-06762-f004:**
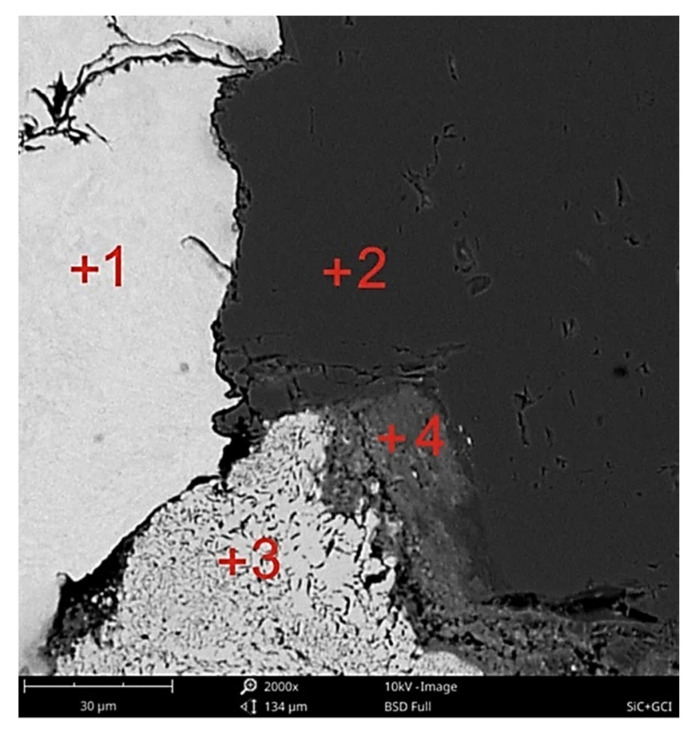
Composite microstructure of the gray cast iron–β-SiC in the area of the matrix contact with the reinforcement containing marked EDS analysis points, SEM, mag. 2000×, nonetched sample; 1—β-SiC reinforcement; 2—metallic matrix (gray cast iron); 3—transition zone 1; 4—transition zone 2.

**Figure 5 materials-14-06762-f005:**
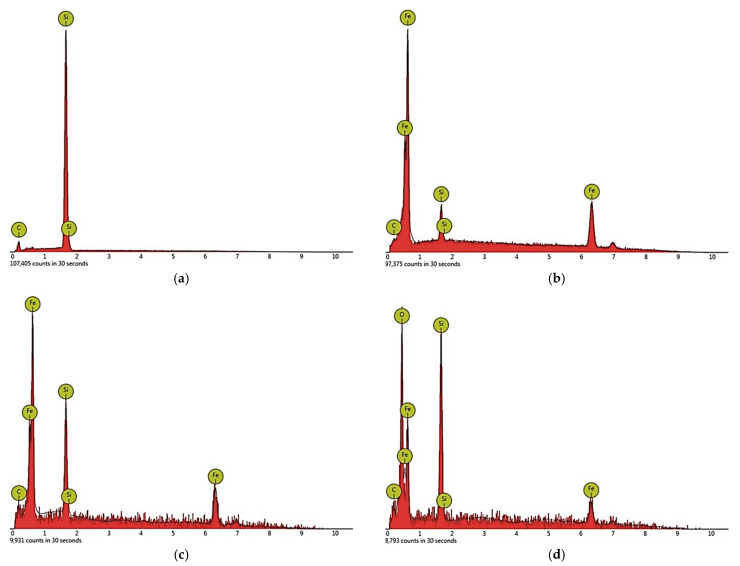
EDS point analysis spectrograms: (**a**) point 1 in Figure 4; (**b**) point 2 in Figure 4; (**c**) point 3 in Figure 4; (**d**) point 4 in Figure 4.

**Figure 6 materials-14-06762-f006:**
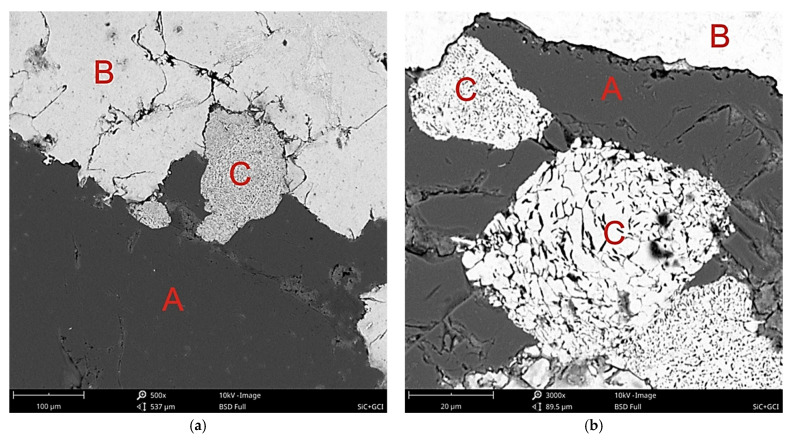
Composite microstructure of gray cast iron–β-SiC in the area of the matrix contact with the reinforcement, SEM; (**a**) mag. 500×; (**b**) mag. 3000×; (**c**) mag. 6000×; (**d**) mag. 6100×, nonetched sample; A—β-SiC reinforcement; B—metallic matrix (gray cast iron); C—transition zone 1.

**Figure 7 materials-14-06762-f007:**
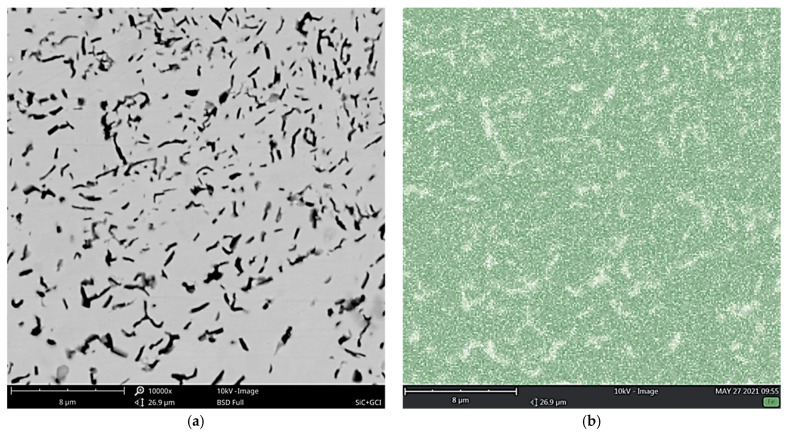
The microstructure of the transition zone 1 and the distribution map EDS of elements in this area: (**a**) tested area, SEM, mag. 10,000×, nonetched sample; (**b**) distribution of Fe at concentrations in the tested area 50.81 %at./77.01 %wt.; (**c**) distribution of Si at concentrations in the tested area 15.93 %at./12.15 %wt.; (**d**) distribution of C at concentrations in the tested area 33.25 %at./10.84 %wt.

**Table 1 materials-14-06762-t001:** Properties of silicon carbide [11].

SiC, %wt.	Fe_2_O_3_, %wt.	C, %wt.	Magnetic Fraction, %wt.	Density, g/cm^3^
Green SiC				
≥97.0	≤1.0	≤0.2	≤0.4	3.1 ÷ 3.2
Black SiC				
≥96.0	≤1.0	≤0.3	≤0.4	3.1 ÷ 3.2
Metallurgical SiC				
≥88.0	≤1.0	≤0.3	≤0.5	3.1 ÷ 3.2

**Table 2 materials-14-06762-t002:** Chemical composition of the metallic matrix: gray cast iron with flake graphite.

C, %wt.	Si, %wt.	Mn, %wt.	Cr, %wt.	Cu, %wt.	Ni, %wt.	P, %wt.	S, %wt.	Fe, %wt.
2.81	1.24	0.61	0.11	0.09	0.05	0.62	0.10	rest

**Table 3 materials-14-06762-t003:** Chemical composition of β-SiC silicon carbide used in the tests according to the manufacturer’s data [15].

SiC, %wt.	C, %wt.	Si, %wt.	SiO_2_, %wt.	Fe_met_, %wt.
≥97.0	≤0.3	≤1.0	≤1.0	≤0.3

**Table 4 materials-14-06762-t004:** The result of the EDS point analysis (marking points according to Figure 4).

C		O		Si		Fe	
%at.	%wt.	%at.	%wt.	%at.	%wt.	%at.	%wt.
Point 1 in Figure 4							
52.03	31.69	-	-	47.97	68.31	-	-
Point 2 in Figure 4							
12.75	3.19	-	-	8.25	4.83	79.00	91.97
Point 3 in Figure 4							
18.49	5.19	-	-	17.94	11.78	63.57	83.02
Point 4 in Figure 4							
14.87	6.19	37.23	20.63	20.23	19.68	27.66	53.50

## Data Availability

The data presented in this study are available in article.

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
