# Peer review of "Surface Phenomena at the Interface between Silicon Carbide and Iron Alloy"

_materials, 2021, doi:10.3390/ma14226762_

Round 1

Reviewer 1 Report

In this work, the authors report the surface phenomena at the interface between silicon carbide and iron alloy, and granular carborundum (β-SiC) is used as a reinforcement. The changes at the interface are discussed in detail, but more necessary characterization is needed. The paper deserves to publish in the Materials after addressing the following comments.

  1. XRD should be used to confirm the phase of the synthesized samples. And it is recommended to supplement TEM and HRTEM test data.
  2. How are the tribological properties of the composite enhanced? Where are the enhanced evidences? Please add relevant performance tests.
  3. Please use pictures with higher resolution to ensure that the content of the pictures can be displayed clearly.
  4. Please pay attention to the layout of the pictures. For example, in Figure 3, a block in the lower right corner is unsightly.
  5. The structures of Tables should be revised to be clearer and easier to read, such as Table 4.
  6. Please pay attention to the expression of the legends. For example, “Figure 6. Composite microstructure of gray cast iron b-SiC in the area of the matrix contact with the reinforcement, SEM, a) mag. 500x, b) mag. 3,000x, c) magnific. 6,000x, c) mag. 6,100x., non-etched sample;”. And please check the other legends carefully.
  7. Please pay attention to some small details in the text. For example, line 14, "carried out by using light (LOM) and scanning electron (SEM) microscopy", so what is LOM? And, "(SEM)" should be placed after "microscopy". Line 205, what does “Figures 2¸7” mean? Please check yourself to determine if there are any other minor errors in the paper.

Author Response

Dear Sir,

Thank you for Your review of our manuscript materials-1428565 entitled "Surface phenomena at the interface between silicon carbide and iron alloy". The manuscript has been modified according to the most comments made by the Reviewers. The additions or modifications made in the manuscript appear in yellow text. Moreover, detailed responses to all Your comments are given in the separated pdf.

Reviewer 2 Report

The paper “Surface phenomena at the interface between silicon carbide and iron alloy” by M. Cholewa et al. reports the synthesis of a composite made by cast iron with flake graphite and ceramic SiC, whose microstructure is studied by microscopy and EDS analysis. The work is clearly described in the abstract, but the article instead suffers of poor clarity and needs more care in the presentation.

I suggest to revise the whole article paying particular attention to the following points:

  1. The introduction section should be rewritten clearly describing what has been already reported about this material in the previous literature and which are the potential applications of the results here reported. The first part about the goals of the work should be summarized in a clearer way. Moreover the authors should avoid the presentation of available literature in the bullet format, but rather describe previous results and how they can be related to the reported work.
  2. The second section should be renamed “materials and methods”
  3. Figures 1, 2 and 5 at least should be presented with a better resolution
  4. It is hard to follow the “Results and Discussion Section”, I suggest to revise it clearly distinguishing between the measurements evidences and the possible explanations and conclusions.
  5. Please also avoid bullet style to present the conclusions. Also specify how possible applications could be verified.

Author Response

(The authors gave the same response as above.)

Round 2

Reviewer 2 Report

The authors addressed my comments. Some deductions, still present in the Research results section, could be shifted to the Discussion section, but the paper clarity has been overall improved. However I warmly suggest to shift the figures 6 and 7 before the Discussion section, since they belong to the previous section.

Author Response

Dear Sir,

Thank you for Your second review of our manuscript materials-1428565 entitled "Surface phenomena at the interface between silicon carbide and iron alloy". The manuscript has been modified according to Your comment. The modifications were made in the manuscript using the “Track Changes” function.

Comments of Reviewer 2 in round 2:

The authors addressed my comments. Some deductions, still present in the Research results section, could be shifted to the Discussion section, but the paper clarity has been overall improved. However I warmly suggest to shift the figures 6 and 7 before the Discussion section, since they belong to the previous section.

Response:

Done. Figures 6 and 7 were shifted to section 3. Research results i.e. before section 4. Discussion of research results.

Best Regards

Authors of paper